# Academic motivation, procrastination, and adjustment: Exploring their impact on student profiles and academic performance

Patra Vlachopanou[1]*, Laura Maska[2], Dimitrios Kalamaras[2], Fani Nasika[2]

1 Aegean College, University of Essex and University of Nicosia, Nicosia, Cyprus, 2 Aegean College, University of Essex, Nicosia, Cyprus

* Vlachopanou.p@unic.ac.cy

## Abstract

### Introduction

Succeeding in entering higher education studies is influenced by motivation, procrastination, and how well students adjust to university life. By understanding these factors, better strategies for supporting students can be developed.

### Aim

This study aims to group university students based on their motivation, procrastination, and adjustment to university, and to examine how these groups relate to their Grade Point Average (GPA).

### Method

284 university students participated in this study, averaging 21.2 years old, of whome 36.2% were male and 63.4% female. Academic motivation was measured using the Academic Motivation Scale, procrastination with the Procrastination Assessment Scale Questionnaire, and adaptation with the Student Adaptation to College Questionnaire. A K-Mean cluster analysis and decision tree methods were used to identify student profiles and their link to GPA.

### Results

Three profiles emerged: (a) Highly Motivated and Well-Adjusted (highest GPAs), (b) Moderately Motivated and Adjusted (average GPAs), and (c) Procrastinated and Poorly Adjusted (lowest GPAs).

**Data availability statement:** All relevant data are within the manuscript and its Supporting Information files.

**Funding:** The author(s) received no specific funding for this work.

**Competing interests:** The authors have declared that no competing interests exist.

## Conclusion

Managing procrastination, staying motivated, and adjusting well to university are key to academic success. Targeted interventions can improve these areas and boost student performance.

## Introduction

Academic adjustment is a multifaceted consept that plays a central role in student success in higher education, encompassing cognitive, behavioral, emotional, and social adaptation to the academic environment [1–4]. It includes the students' capacity to meet academic challenges, integrate into the social context, and manage the psychological pressures of university life. Inadequate academic adjustment has been associated with increased dropout rates, mental health difficulties, and lower academic performance, particularly in transitional periods such as the first year of university [5–6].

Despite the growing body of literature on academic adjustment, several limitations remain. Prior research tends to focus on isolated factors such as motivation or procrastination in silos, without accounting for their dynamic interplay in shaping academic adaptation [7–8]. Moreover, much of the existing work lacks culturally contextualized insights, especially within Southern European settings like Greece, where systemic and socio-economic stressors may uniquely affect student adjustment [9]. This study aims to address this literature gap by exploring integrative profiles of students that include motivational and behavioral dimensions to predict academic adaptation and performance.

*In Greece, academic adjustment among university students is often challenged by a combination of structural and cultural stressors* [10–13] These include limited academic counseling services, high levels of youth unemployment, and a pervasive societal expectation to pursue higher education regardless of personal interest or readiness [14]. These factors may contribute to an increase in amotivation and procrastination among students who often enter university without clear vocational orientation or internalized goals [15]. Although Greek students demonstrate high rates of enrollment, dropout rates and mental health concerns remain persistent issues [16]. Thus, studying academic motivation, procrastination, and adjustment in this context provides both theoretical and practical insights for culturally informed interventions.

Academic adjustment has been consistently associated with better academic outcomes, greater university satisfaction, intrinsic motivation, and psychological well-being [2–4,6]. Students with higher adjustment scores tend to navigate the challenges of higher education more successfully, showing higher resilience and better social integration [17]. Conversely, students with poor adjustment face academic stress, disengagement, and social withdrawal.

One key factor affecting adjustment is academic procrastination. It refers to the voluntary and irrational delay of academic tasks despite awareness of potential negative consequences [18]. It has been shown to undermine students' academic

performance, cause stress, and reduce well-being [19]. Procrastination has also been linked to self-regulatory deficits, a poor sense of efficacy, and weak time-management skills-factors that hamper smooth academic integration [20].

In parallel, academic motivation-conceptualized as intrinsic motivation (IM), extrinsic motivation (EM), and amotivation-plays a foundational role in determining how students engage with their studies [21]. Intrinsic motivation involves engagement for inherent interest or enjoyment and is consistently related to deep learning and high achievement [22]. More recently, longitudinal studies have reaffirmed that intrinsic motivation predicts long-term academic persistence and meaningful engagement [23].

Extrinsic motivation, driven by external rewards or the avoidance of punishment, shows a more nuanced relationship with academic adjustment. While it may be effective in initiating academic behaviors, it is less likely to sustain engagement over time or foster deep learning [24–25]. The temporary benefits of extrinsic motivation can be overshadowed by anxiety and surface-level learning strategies.

Amotivation, the state of lacking intent or purpose to act, is perhaps the most detrimental motivational profile. It is associated with disengagement, increased procrastination, and difficulties in adjusting to academic demands [8, 26]. Students with high amotivation often perceive academic demands as meaningless or unmanageable, leading to psychological distress and reduced performance.

Despite growing recognition of these psychological and behavioral dimensions, there remains a dearth of studies that examine the interrelation between academic motivation, procrastination, and adjustment using person-centered approaches such as clustering [27]. This methodological gap limits our understanding of how student profiles influence academic outcomes in real-world settings.

Furthermore, little attention has been given to how these variables predict performance across diverse academic cultures. This study aims to fill this research gap by investigating the motivational and procrastinatory profiles of Greek university students and their associations with academic adjustment and GPA.

## Rationale and research gaps

The core motivation for the present research lies in the recognition that academic success is a multidetermined outcome, influenced by the intersection of motivation, behavioral regulation, and adjustment [28]. Existing models often rely on variable-centered analyses, which may obscure how real-world students operate within multidimensional psychological profiles [29].

To address this, the present study adopts a person-centered approach (cluster analysis), grouping students based on shared levels of motivation and procrastination. This method allows for identifying nuanced patterns of behavior and provides a more ecologically valid framework for intervention. Additionally, by focusing on a Southern European context-where socio-cultural and economic stressors uniquely affect students-this study offers a localized contribution to a globally relevant issue [30].

Importantly, this study also responds to the lack of methodological clarity in previous research. Many prior studies do not adequately explain the robustness of their models. In contrast, the current study employs a data-driven clustering approach combined with ANOVA and post-hoc testing to ensure replicability and analytical transparency. This methodology allows for a more detailed understanding of the underlying student profiles that affect academic success [31].

To enhance transparency and reproducibility, a sample screening process flow diagram is proposed, outlining participant inclusion, exclusion, and final sample size (see Fig 2).

In addition, this study contributes to literature by bridging gaps between theoretical concepts and applied outcomes. While motivation, procrastination, and adjustment have been individually studied, few studies have holistically examined how these factors co-occur within the same individuals using a person-centered lens [32, 4]. Moreover, this study moves beyond simply confirming known relationships by identifying actionable profiles with practical implications for educators

and policymakers. Thus, the originality of the present work lies not in isolating novel consepts, but in synthesizing well-established ones into an integrative, empirically grounded typology that supports targeted intervention strategies.

## Hypotheses

Hypothesis 1: Three distinct motivational–procrastination clusters are expected to emerge.
Hypothesis 2: Cluster 1 will comprise students with low procrastination, high intrinsic motivation, and low amotivation. These students are hypothesized to demonstrate the highest academic adjustment and GPA.
Hypothesis 3: Cluster 2 is expected to include students with moderate intrinsic motivation, moderate procrastination, and low amotivation. They are anticipated to show average levels of academic adjustment and GPA.
Hypothesis 4: Cluster 3 is predicted to consist of students with high procrastination, low intrinsic motivation, and high amotivation, resulting in the lowest academic adjustment and GPA.

## Method

### Participants

This is a survey study, based on a correlational research design between participants, designed to establish a relationship between specific individual characteristics of students (independent variables) and the students' academic performance (dependent variable). The study involved 284 participants aged 18–28 years (*M*: 21.2 years; *SD* = 1.7). The sample comprised 36.2% male and 63.4% female students (see Table 1).

The participants are mostly from second-, third-, and fourth-year students (33.9%, 20.5%, and 25.8% respectively). Freshmen make up 13.8% of the sample. Additionally, it is worth noting that a small percentage of participants (6.0%) are students who have exceeded the fourth year of study, which is the minimum time required to obtain their degree.

The majority of students chose Psychology (127 students, 46.2%), followed by Informatics and Telecommunications (63 students, 22.9%) and Nursing (53 students, 19.3%). The Education departments gathered 24 students (8.7%), while smaller percentages appeared in Theology (13, 4.7%), Life and Health Sciences (12, 4.4%), Engineering/Physical Sciences (6, 2.2%), and Social Sciences/Humanities (5, 1.8%). While the Mathematics, Business Administration and Agricultural Studies departments are underrepresented with percentages below 1.1% for each department.

A significant majority of the participants (79.2%) reported that their hometown was different from their university location. Nearly half of the participants (48.6%) indicated that their current university department was their top choice. Additionally, 75.4% of the respondents expressed satisfaction with both the teaching methods and the curriculum.

### Recruitment period

Participants were recruited between January 19th, 2024, and April 26th, 2024

### Procedure

Undergraduate students from universities in Attica and other regions participated in the study by completing online questionnaires via Microsoft Survey Form, organized with the support of university administrations. The process was anonymous and voluntary, adhering (see Fig 1).

### Ethics and informed consent

Ethical approval for the study was obtained from the Research Ethics Committee of the Postgraduate Research Programme at Aegean Omiros College (PGR AOC) on 16/10/2023 (Approval Code: PGR-AOC-2023-10PD23).

**Table 1. Participants' Characteristics and descriptive statistics (N = 284).**

| Variables | Values and Coding | Percentage/Average |
|---|---|---|
| Age | Ranging 18-28 | 21.2 years (SD= 1.7) |
| Gender | 1, Male | 38.0% |
| | 0, Female | 62.0% |
| Year of study | 1st | 13.8% |
| | 2nd | 33.9% |
| | 3rd | 20.5% |
| | 4th | 25.8% |
| | 5th and above | 6.0% |
| Department of study | Psychology | 46.2% |
| | Informatics and Telecommunications | 22.9% |
| | Nursing | 19.3% |
| | Education | 8.7% |
| | Theology | 4.7% |
| | Life and Health Sciences | 4.4% |
| | Engineering/Physical Sciences | 2.2% |
| | Social Sciences/Humanities | 1.8% |
| | Mathematics, Business Administration and Agricultural Studies | below 1.1% |
| Hometown location is same as university location | 1, Yes | 20.8% |
| | 0, No | 79.2% |
| The enrolled department was my first choice | 1, Yes | 48.6% |
| | 0, No | 51.4% |
| Grade point average (GPA) | Scale [0-10] | 7.56 (SD= 0.95) |
| Satisfied with the way of teaching and the curriculum | 1, Yes | 75.4% |
| | 0, No | 24.6% |

All participants provided informed, written consent prior to participation. Consent was obtained electronically through the survey platform, and completion of the consent form was a prerequisite for accessing the questionnaires. As such, participants could not proceed without first agreeing to the terms of participation.

The study did not involve minors. Participant age eligibility was clearly indicated in the study title and description prior to consent.

## Measures

### Academic motivation scale

Academic motivation was assessed using the Academic Motivation Scale (AMS, AMS-C-28) [33], which includes 28 items rated on a Likert scale. The Greek adaptation of the AMS by [34] was used, featuring seven subscales that measure three types of intrinsic motivation (to know, to accomplish, and to experience stimulation), three types of extrinsic motivation (external regulation, introjected regulation, and identified regulation), and amotivation. Responses are rated from 0 (not at all) to 7 (exactly). Higher scores in each subcategory reflect stronger motivation. The reliability of the scale is high, with a Cronbach's $\alpha$ of .89 for intrinsic motivation, .73 for extrinsic motivation, and .86 for amotivation.

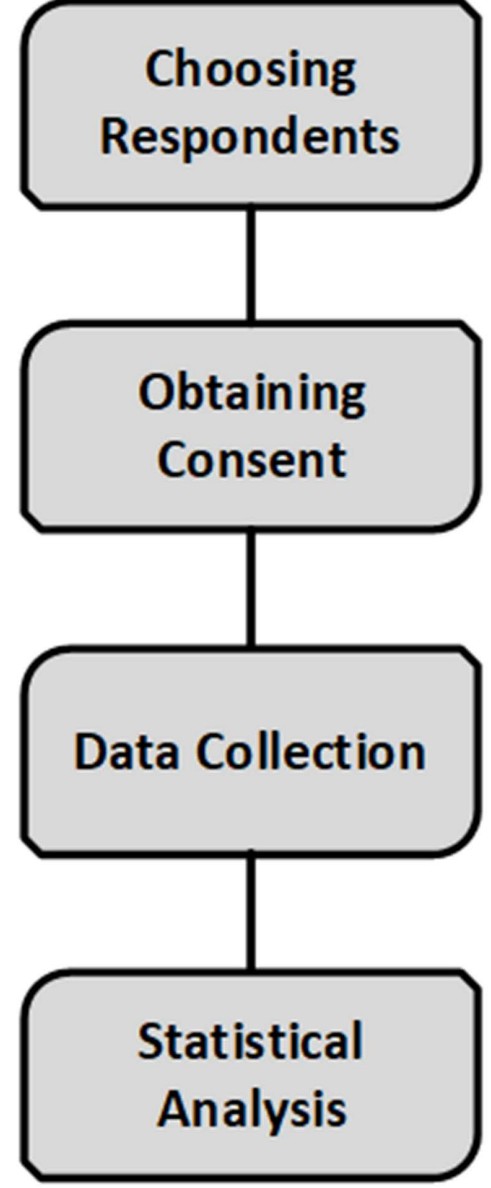

**Fig 1. Flow chart of the study procedure.**

## Procrastination assessment scale questionnaire

Procrastination was measured using the Procrastination Assessment Scale Questionnaire (PASS) [35], adapted into Greek (Chatzidimou, S. V. [Unpublished]). This self-report measure evaluates procrastination behaviors across five academic tasks: writing term papers, studying for exams, handling academic-administrative tasks, attending classes, and general school activities. Responses are recorded on a 5-point Likert scale, from 1 (never procrastinate) to 5 (always procrastinate). For this study, only the initial questions from each subscale were utilized, as recommended by [2–4,36,37]. The scale showed a Cronbach's $\alpha$ of .88.

## Student adaptation to college questionnaire

The Student Adaptation to College Questionnaire [38], translated and adapted into Greek by [39], was used to measure adaptation. This 67-item self-report questionnaire employs a 9-point Likert scale, ranging from 1 (strongly disagree) to 9 (strongly agree). It assesses four dimensions of adaptation: academic, institutional, social, and personal-emotional. Participants reflected on their most recent university experiences. The scale's reliability is high, with a Cronbach's *a* of .93.

## Statistical analysis

As mentioned in the introduction section, this study sought to cluster the variables of academic motivation, academic volatility and academic adjustment to examine a) whether there are distinct student profiles that explain students' adjustment to university life and b) which of these profiles are associated with higher GPA scores. To do that, the first correlations were performed among the variables under investigation (i.e., academic motivation, academic volatility, academic adjustment and the students' academic year). Furthermore, standardized scores for the above-mentioned variables were entered into a K-Mean cluster analysis algorithm using Ward's method to identify distinct student profiles regarding their defense styles of academic motivation, academic volatility and academic adjustment. Furthermore, a decision tree approach was used to model the relationship between the identified student profiles (from K-means clustering) and academic performance (GPA). The decision tree method, specifically CHAID, was chosen for its interpretability and ability to handle both categorical and continuous variables. It allows for the exploration of how combinations of variables (e.g., cluster membership and academic year) predict grade point average (GPA) in an easily visualized format. This approach is particularly useful for identifying key decision rules and risk groups, supporting practical educational interventions.

The choice of K-means and decision tree clustering methods (CART with CHAID) in this study was strategic and appropriate, given the objectives and data structure. K-mean algorithm, due to its computational efficiency, simplicity, and interpretability, makes it particularly suitable when the goal is to explore clear, non-overlapping clustering in a moderately sized data set, as was the case in this study (N = 284). In summary, K-means was chosen over alternative clustering methods due to its methodological alignment with the study objectives, its effectiveness in handling continuous psychological variables, and its theoretical compatibility with pre-established hypotheses regarding the structure of student profiles. In terms of fit to context, both methods were appropriate for the exploratory nature of the study. Cluster analysis revealed significant profiles with a variety of academic behaviors, while decision trees clearly demonstrated how these profiles are related to academic outcomes.

Additionally, the already mentioned clustering solution was evaluated by considering 24 criteria (see Appendix). At a next step, multivariate analysis of variance (MANOVA) was used to explore the differences among clusters, whereas a decision tree–CART analysis was used with the chi-squared automatic interaction detection (CHAID) method to assess the effect of student profile and year of study on their performance. The analyses were performed using R statistical software and the Statistical Package for Social Sciences software (SPSS, version 29.0), respectively.

# Results

## Correlation analysis

As expected, procrastination assessment scale was negatively correlated to intrinsic motivation ($r = -.48$, $p < .001$) and adaptation to college ($r = -.56$, $p < .001$) and positively correlated to extrinsic motivation ($r = .23$, $p < .001$) and amotivation ($r = .20$, $p < .001$). Intrinsic motivation was positively correlated to adaptation to university ($r = .52$, $p < .001$) and negatively correlated to procrastination assessment ($r = -.48$, $p < .001$) and amotivation ($r = -.31$, $p < .001$) while extrinsic motivation was positively correlated to the procrastination assessment scale ($r = .23$, $p < .001$). Amotivation was positively correlated to procrastination ($r = .20$, $p < .001$) and negatively correlated to academic intrinsic motivation ($r = -.31$, $p < .001$) and adaptation to university ($r = -.46$, $p < .001$). Finally, student adaptation to university was positively correlated to intrinsic motivation ($r = .52$, $p < .001$) and negatively correlated to procrastination ($r = -.56$, $p < .001$) and amotivation ($r = -.46$, $p < .001$).

## Cluster analysis

Cluster analysis produced a three-cluster solution that included three unique student profiles. The first profile included "Highly Motivated and Well-Adjusted Students", the second profile included "Moderately Motivated and Adjusted Students", whereas the third student profile included the "Procrastinated and Poorly Adjusted Students". The means of subscales for each cluster, the standard deviation, and the *p*-values for each cluster can be found in Table 3.

Subsequently, a MANOVA model was used to examine differences across the variables of academic motivation, academic procrastination, and academic adjustment among student profiles Table 2 shows the results of the MANOVA examination on how the student profiles and year of study differ regarding the above variables. As shown in Table 2, only the student profiles ($p < 0.001$) are statistically significantly correlated with the coverage of the indicators under study (all together, with the variables reflecting academic motivation, academic procrastination, and academic adjustment).

Based on multiple comparisons results (see Table 3.), it is argued that students in the Cluster 3 (Procrastinated and Poorly Adjusted Students) reported the highest levels on the Procrastination Assessment Scale of participation, compared to the other Clusters, as Cluster 1 (Highly Motivated and Well-Adjusted Students) differed clearly from the other two Clusters, with the highest score on Intrinsic motivation. Regarding Extrinsic motivation and amotivation, it seems that students belonging to Cluster 3 tend to be more externally motivated and unmotivated, in comparison to others (belonging to Custer 1 or Cluster 2). Finally, students belonging to Cluster 1 and Cluster 2 (Highly Motivated and Well-Adjusted Students and Moderately Motivated and Adjusted Students) tend to be more adapted to the university environment in comparison to Cluster 3 students. In other words, the three Clusters represent three different student profiles in meaningful ways on the criteria variables.

## Decision tree model

To examine which of these profiles are associated with higher GPA scores, a decision tree–CART analysis mode was applied with GPA as the outcome variable (exhaustive CHAID was the growing method). Both student profiles (cluster

**Table 2. Results of a 4×2 MANOVA examining differences of student profiles and year of study on measures used in the cluster analysis (z-scores).**

| MANOVA | | Value | F | Sig. |
|---|---|---|---|---|
| Students' profile | Pillai's Trace | .693 | 27.597 | .000 |
| | Wilks' Lambda | .318 | 40.026[a] | .000 |
| Year of study | Pillai's Trace | .142 | .958 | .546 |
| | Wilks' Lambda | .864 | .964 | .536 |

**Table 3. Underlying student profiles regarding academic motivation, academic procrastination, and academic adjustment.**

| Mean (SD) | Cluster 1 (n=75) | Cluster 2 (n=107) | Cluster 3 (n=101) | *Mean Diference[a]* | *Mean Diference[b]* | *Mean Diference[c]* |
|---|---|---|---|---|---|---|
| Procrastination Assessment Scale for Students·(PASS)· | 9.08 (3.3) | 12 (3.62) | 15.53 (4.68) | **−2.92\*** | **−6.36\*** | **−3.44\*** |
| Intrinsic motivation – to know | 6.07 (0.84) | 5.44 (1.22) | 3.92 (2.08) | **.63\*** | **2.12\*\*** | **1.49\*** |
| Extrinsic motivation – identified | 5.27 (1.52) | 5.05 (1.44) | 5.74 (1.52) | .22 | −.45 | **−.67\*\*** |
| Amotivation | 1.19 (0.46) | 1.38 (0.65) | 2.13 (1.65) | −.19 | **−.9184\*** | **−.7317\*** |
| Student Adaptation to College (SACQ total score) | 458.63 (25.88) | 391.97 (20.06) | 310.81 (33.38) | **66.66\*** | **147.94\*** | **81.28\*** |

Note: The statistically significant differences are presented with bold ($p < 0.001$; \*\*$p < 0.05$; \*\*\*$p < 0.10$). Subscripts (a, b, c) in each column denote statistically significant difference in post-hoc multiple comparisons, using Bonferroni correction as follows: a = Difference between the first and the second cluster, b = Difference between the first and the third cluster, c = Difference between the second and the third cluster. Cluster 1 = "Highly Motivated and Well-Adjusted Students", Cluster 2 = "Moderately Motivated and Adjusted Students", and Cluster 3 = "Procrastinated and Poorly Adjusted Students".

membership variable) and academic year have been included in the model, although no statistically significant connection was found between the academic year on measures used in the cluster analysis (*see* Table 2).

The analysis revealed meaningful patterns in the relationship between student motivation, adjustment, and academic performance. As anticipated, distinct behavioral and motivational profiles emerged across the three identified clusters, each demonstrating varying levels of academic success. These findings offer empirical support for the hypothesis that higher intrinsic motivation and better personal adjustment are positively associated with improved academic outcomes.

It is clear from these results (see Fig 2) that Cluster 1 had a better mean score in comparison with the other two Clusters (*M*=7.981; *SD*=.830). This is like saying that Highly Motivated and Well-Adjusted Students ensure better academic performance than others. Cluster 2 (Moderately Motivated and Adjusted Students) had the second highest mean (*M*=7.701; *SD*=.805), followed by Cluster 3 (*M*=7.100; *SD*=.987), whose members represented the Procrastinated and Poorly Adjusted Students, who seem to have the worst academic performance compared to all other students.

## Discussion

This study intended to explore how academic motivation, procrastination, and adjustment interplay to explain student outcomes, and more specifically GPA. Although a combination of these variables has been documented in the literature to create associations, further research on how they can interact was needed [2,3,4,40,41]. Using cluster analysis, this study classified students into groups based on their levels of intrinsic motivation, extrinsic motivation, and amotivation and aimed to understand academic procrastination in each group as well as the lack of student adjustment to university. Tree analysis also allowed us to determine exact motivational profiles that are associated with successful university adaptation and provided a more complete picture of how students adapt in higher education.
Three profiles emerged from the cluster analysis

### Profile one: Highly motivated and well-adjusted students

This profile includes students demonstrating high intrinsic motivation and low academic procrastination. These characteristics predict successful outcomes in both adaptive university adjustment and GPA. This aligns with research by [42] which showed that intrinsic motivation is associated with better adjustment to university life [43]. Moreover, the study by [2] has shown that intrinsic motivation decreases procrastination, leading to a positive effect on academic adjustment. [44] also suggested that these students are more likely to do well academically because of their intrinsic motivation and strong self-regulation.

### Profile two: Moderately motivated and adjusted students

This group, labelled as "Innovative Students," is characterized by a strong sense of external motivation, such as the drive to achieve good grades or secure future job opportunities, paired with a moderate level of internal motivation. They are not as intensely driven as the top-performing group, but their external goals seem to play a significant role in helping them adjust to university life. These students manage to balance their internal motivation with a tendency to procrastinate, which can sometimes interfere with their adjustment and academic performance. Research shows that, while external motivation can aid in university adjustment, it may also lead to procrastination [44,45]. Innovative Students enter higher education with clear goals and follow established procedures to meet the minimum requirements for adapting to their new environment. This approach typically ensures they complete their studies within a reasonable timeframe, but their procrastination habits might occasionally affect their overall experience and performance [46,47].

### Profile three: Procrastinated and poorly adjusted students

The third profile includes students with low intrinsic motivation and high levels of procrastination. These students lack the internal drive to engage deeply with their academic work and are prone to delaying tasks [48]. Their high levels of

# Grade point average (GPA)

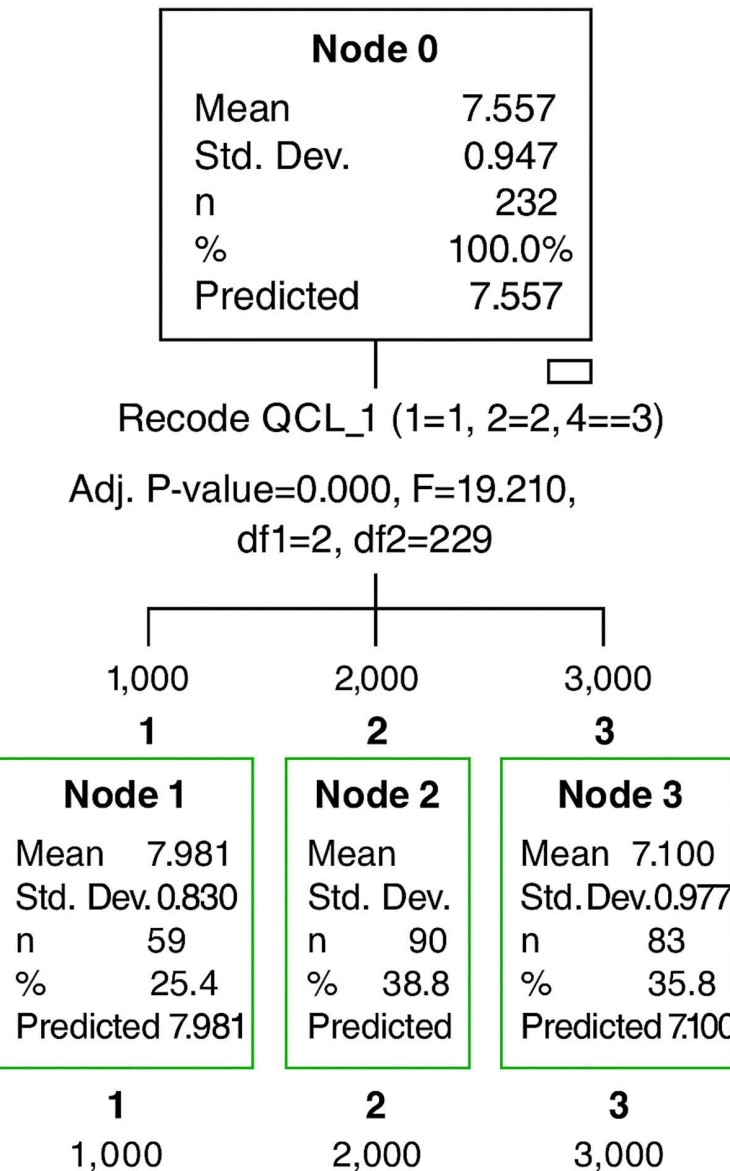

**Fig 2. Results concerning decision tree model with GPA as an outcome variable.**

procrastination significantly impair their academic adjustment and performance [48–50] found that amotivation leads to disengagement, higher procrastination, and lower academic adjustment, which aligns with this profile. These students struggle academically, highlighting the detrimental effects of low intrinsic motivation and high procrastination on academic success [12].

As indicated above, all profiles show high levels of amotivation. This can be attributed to a long-running cultural narrative in Greece, where if someone finishes high school and does not immediately pursue further education, this is

perceived as a failure. Consequently, students enter higher education with varying levels of intrinsic and extrinsic motivation, influenced by their parents and societal pressures [23].

These results align with general research on student motivation and performance [51]. Prior research has established a persuasive negative relationship between academic procrastination and achievement in school. High procrastinators score lower on objective measures of GPA [48,52]. The results of the present study corroborate these findings, as the Procrastinated and Poorly Adjusted Students profile was associated with lower GPAs, while the Highly Motivated and Well-Adjusted Students were associated with the highest GPAs.

Moreover, intrinsic motivation has been extensively linked to better academic outcomes [40]. The analysis of the present study supports this, showing that the Highly Motivated and Well-Adjusted Students, who exhibited the highest intrinsic motivation, also achieved the highest GPAs. In contrast, the Procrastinated and Poorly Adjusted Students, characterized by low intrinsic motivation, corresponded to the lowest GPAs. This underscores the importance of fostering a genuine interest in academic material to enhance student performance.

The role of extrinsic motivation in this study also offers insightful implications. Although extrinsic motivation did not vary drastically across profiles, it was highest in the Procrastinated and Poorly Adjusted Students and lowest in the Moderately Motivated and Adjusted Students. Previous studies suggest that while extrinsic motivation can drive performance, it may not be as effective as intrinsic motivation in promoting long-term academic success [53]. The findings of the present study suggest that the higher levels of extrinsic motivation in the Procrastinated and Poorly Adjusted Students did not compensate for their low intrinsic motivation and high procrastination, resulting in lower academic achievement.

Amotivation, or the lack of motivation, further delineated the profiles, with the Procrastinated and Poorly Adjusted Students having the highest levels of amotivation. This is significant, as amotivation has been negatively associated with academic outcomes [54]. The high amotivation in this group could explain their poor academic performance, underscoring the need for interventions aimed at reducing amotivation among students.

Lastly, this study confirms the critical role of student adaptation to university life. The Highly Motivated and Well-Adjusted Students displayed higher adaptation levels, which corresponded to better academic performance. This resonates with previous research indicating that well-adapted students are more likely to succeed academically [55]. In contrast, the lower adaptation scores in the Procrastinated and Poorly Adjusted Students align with their poorer academic outcomes, emphasizing the importance of supporting students' transition to university life [56,57].

Practical applications of these findings should be emphasized more explicitly. Universities and educational institutions could use this evidence to design targeted intervention programs, such as workshops to enhance intrinsic motivation, time management training to reduce procrastination, and mentoring schemes to support student adjustment [57]. For instance, implementing self-regulation and goal-setting modules into first-year orientation programs could preemptively address these issues. Academic advising practices may also be revised to include screening for procrastination tendencies and tailoring support accordingly [58].

Furthermore, while this study provides valuable insights into motivational profiles, the inclusion of additional variables such as socio-economic status (SES), mental health indicators (e.g., anxiety or depression), and family background may yield a more nuanced understanding of the interplay between motivation, procrastination, and adjustment. These factors often exert a significant influence on students' academic trajectories and could further refine support strategies aimed at improving outcomes. Future research should aim to incorporate such dimensions to enhance the generalizability and applicability of findings.

Finally, considering the specific cultural and educational context in Greece is essential for fully understanding the dynamics observed in this study. Greek educational culture often emphasizes academic achievement as a primary measure of personal success, heavily influenced by familial expectations and rigid societal norms [59]. The prevalence of extrinsically motivated behavior and amotivation among students may stem from this pressure-laden environment, where

entering university is treated as an obligation rather than a personal choice. Moreover, the centralized and exam-oriented nature of the Greek educational system may not sufficiently cultivate self-regulated learning, thereby reinforcing procrastination and hindering student adjustment. These culturally embedded patterns highlight the need for systemic educational reforms and culturally responsive interventions tailored to the Greek context [60].

## Conclusion

In conclusion, the cluster and tree analyses underscore the multifaceted nature of academic performance, where low procrastination, high intrinsic motivation, and effective adaptation to university life are key predictors of academic success. These findings align with existing research and suggest potential areas for targeted interventions to improve student outcomes.

Beyond school-level interventions, such as mentoring programs, academic counseling, and time management workshops, these results also hold important implications for policymakers. Educational policy should prioritize early identification systems for at-risk students, integrating psychological assessments of motivation and self-regulation into the university enrollment or orientation process. Additionally, national education strategies can promote funding for institutional support services, ensuring universities are adequately resourced to address student adjustment challenges. Policies that incentivize training faculty and advisors in motivational interviewing or supportive academic coaching may further enhance student engagement and success.

Thus, a dual approach is recommended: institution-level interventions that provide immediate, personalized support, and broader policy-level frameworks that ensure systemic, long-term improvements to student well-being and academic integration.

## Limitations

A limitation of the current research design is its reliance on a cross-sectional approach rather than a longitudinal one. Utilizing a longitudinal design would allow for the tracking of variables over time, enabling the analysis of long-term impacts and changes [61]. Another limitation is that the students' grades were self-reported rather than obtained from official departmental records, which may introduce some bias.

## Future studies

Further studies could explore the specific mechanisms by which these factors interact and develop strategies to enhance intrinsic motivation and adaptation skills among students. Specifically, future research could investigate the effectiveness of academic advising, counselling, and mentorship programs in improving student adaptation to university. These services can provide essential support and help students manage academic and personal challenges, thereby enhancing their overall adaptation and academic performance.

## Supporting information

**S1 File. Supporting Information including Appendix (Cluster Evaluation Criteria), raw dataset used for statistical analysis, and CodeBook detailing variable definitions and coding schemes.**
(ZIP)

## Author contributions

**Conceptualization:** Patra Vlachopanou, Laura Maska, Dimitrios Kalamaras.

**Data curation:** Patra Vlachopanou, Dimitrios Kalamaras.

**Formal analysis:** Patra Vlachopanou, Dimitrios Kalamaras.

**Methodology:** Patra Vlachopanou, Dimitrios Kalamaras.

**Supervision:** Patra Vlachopanou, Laura Maska, Fani Nasika.

**Writing – original draft:** Patra Vlachopanou, Laura Maska, Dimitrios Kalamaras.

**Writing – review & editing:** Patra Vlachopanou, Laura Maska, Fani Nasika.

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
