## [Decision Letter · Decision Letter 0]

4 Jun 2025

Dear Dr. Kalamaras,

We look forward to receiving your revised manuscript.

Kind regards,

Assoc. Prof. Phakkharawat Sittiprapaporn, Ph.D.

Academic Editor

PLOS ONE

**Journal Requirements:**

Please ensure that your manuscript meets PLOS ONE's style requirements, including those for file naming. The PLOS ONE style templates can be found at https://journals.plos.org/plosone/s/file?id=wjVg/PLOSOne_formatting_sample_main_body.pdf and https://journals.plos.org/plosone/s/file?id=ba62/PLOSOne_formatting_sample_title_authors_affiliations.pdf 2. We note that you have indicated that there are restrictions to data sharing for this study. For studies involving human research participant data or other sensitive data, we encourage authors to share de-identified or anonymized data. However, when data cannot be publicly shared for ethical reasons, we allow authors to make their data sets available upon request. For information on unacceptable data access restrictions, please see http://journals.plos.org/plosone/s/data-availability#loc-unacceptable-data-access-restrictions.  Before we proceed with your manuscript, please address the following prompts: a) If there are ethical or legal restrictions on sharing a de-identified data set, please explain them in detail (e.g., data contain potentially identifying or sensitive patient information, data are owned by a third-party organization, etc.) and who has imposed them (e.g., a Research Ethics Committee or Institutional Review Board, etc.). Please also provide contact information for a data access committee, ethics committee, or other institutional body to which data requests may be sent. b) If there are no restrictions, please upload the minimal anonymized data set necessary to replicate your study findings to a stable, public repository and provide us with the relevant URLs, DOIs, or accession numbers. Please see http://www.bmj.com/content/340/bmj.c181.long for guidelines on how to de-identify and prepare clinical data for publication. For a list of recommended repositories, please see https://journals.plos.org/plosone/s/recommended-repositories. You also have the option of uploading the data as Supporting Information files, but we would recommend depositing data directly to a data repository if possible. Please update your Data Availability statement in the submission form accordingly. 3. We note that you have referenced “Chatzidimou, S. V.” which has currently not yet been accepted for publication. Please remove this from your References and amend this to state in the body of your manuscript: (Chatzidimou, S. V. [Unpublished]) as detailed online in our guide for authorshttp://journals.plos.org/plosone/s/submission-guidelines#loc-reference-style 4. Please upload a new copy of Figure 1 as the detail is not clear. Please follow the link for more information: https://blogs.plos.org/plos/2019/06/looking-good-tips-for-creating-your-plos-figures-graphics/"
https://blogs.plos.org/plos/2019/06/looking-good-tips-for-creating-your-plos-figures-graphics/

Reviewers' comments:

Reviewer's Responses to Questions

**Comments to the Author**

1. Is the manuscript technically sound, and do the data support the conclusions?

Reviewer #1: Yes

Reviewer #2: Yes

Reviewer #3: Yes

Reviewer #4: Partly

Reviewer #5: Yes

2. Has the statistical analysis been performed appropriately and rigorously?

Reviewer #1: No

Reviewer #2: Yes

Reviewer #3: Yes

Reviewer #4: Yes

Reviewer #5: Yes

3. Have the authors made all data underlying the findings in their manuscript fully available?

Reviewer #1: Yes

Reviewer #2: Yes

Reviewer #3: Yes

Reviewer #4: No

Reviewer #5: Yes

4. Is the manuscript presented in an intelligible fashion and written in standard English?

Reviewer #1: No

Reviewer #2: Yes

Reviewer #3: Yes

Reviewer #4: No

Reviewer #5: Yes

**Reviewer #1:**  The work presented by these authors is very interesting, but here are some limitations, in my opinion. The introduction is not very engaging, and the authors do not clearly present the limitations of the literature on the paper, nor do they address the gap that their work fills. There is no debate raised regarding the directions of the literature concerning new fields of research on the determinants.

If possible, please consider including a flow diagram for the sample screening process to facilitate better understanding of the sampling procedure. In terms of methodology, the authors do not explain the benefits of using a model and the gaps it addresses, as well as the potential robustness of the results compared to other studies.

**Reviewer #2: ** Strengths:

The study employs an appropriate combination of cluster analysis and MANOVA to identify and validate student profiles based on motivation, procrastination, and adjustment.

The table clearly demonstrates meaningful differences between clusters, especially in terms of procrastination and intrinsic motivation.

The profiles generated align with GPA performance, making the results potentially useful for targeted educational interventions.

Suggestions for improvement:

Please clarify how the number of clusters was determined (e.g., elbow method, silhouette analysis, or decision tree splits).

It would strengthen the findings to elaborate on why extrinsic motivation shows no significant differences across profiles, and whether other subdimensions of motivation were considered.

**Reviewer #3: ** The manuscript is generally well-executed, but I have the following suggestions:

Clarify Methodological Choices: It would be helpful to explain in more detail why the K-means and decision tree methods were selected over other potential techniques and the limitations associated with these methods.

Contextualization of Findings: Further exploration of how the results relate to the specific cultural and educational context in Greece would strengthen the manuscript, especially considering the unique influences of Greek educational and societal norms on student behavior.

Practical Applications: It would be beneficial to expand on how universities and educational programs can apply these findings in practical terms to support students better, focusing on interventions that target academic procrastination and motivation.

Consideration of Additional Variables: The study focuses on motivation, procrastination, and adjustment. Including additional variables such as socio-economic status or mental health might provide more comprehensive insights into student success.

**Reviewer #4:**  Higlights

This research is a great contribution to the field of educational psychology. Although the study focuses on variables that have been extensively studied such as academic adjustment, procrastination, and extrensic and intrinsic motivation, it provides a comprehensive discussion on the dynamic interplay of variables under study in Greeks’ context. Discussion of findings flows logically from the study results.

To enhance the paper for publication, the researchers may consider the following:

1. Provide additional studies in Greece on similar variables to illustrate the study’s context. Only one study in Greece focusing on the positive effect of academic adjustment on students’ academic success was included. It will enrich the study’s context if the researcher will provide additional studies along this topic.

2. Revise the introduction to improve its coherence and cohesiveness. The researcher may also include a brief background or context on the status of Greek college students’ academic adjustment, motivation and procrastination in general.

3. Include participants’ relevant demographics such as their year level in college and degree programs because these data can also provide insights to the study’s results.

4. Add the title, Results, after Statistical Analysis to provide demarcation between Methodology and Results Sections.

5. Elaborate on specific interventions recommended in the conclusion section. Would it be school interventions only? Can policy makers be included too?

6. Ensure consistency in writing reference entries, capitalization of journal titles, use of period before the url. Observe hanging indentation.

7. The paper provides interesting, relevant, and insightful findings. To improve its clarity, consider improving awkward sentence construction and cohesion.

**Reviewer #5:**  It is good that the manuscript recognizes the need for academia to remain diligent in seeking ways to continuously improve support of student outcomes.

However, in reading the manuscript I struggle to identify the meaningful contribution or expansion of knowledge. The constructs, academic motivation, procrastination, and adjustment have been extensively studied. Many of the findings are readily available and known.

Perhaps a more beneficial study could assess the influence of online modalities or select AI policies in relation to academic motivation, procrastination, and adjustment.

**Do you want your identity to be public for this peer review?** For information about this choice, including consent withdrawal, please see our Privacy Policy

Reviewer #1: **Yes: ** Shibiru Jabessa Dugasa

Reviewer #2: No

Reviewer #3: No

Reviewer #4: No

Reviewer #5: No

---

## [Author Response · Author response to Decision Letter 1]

8 Jul 2025

Reviewer #1: The work presented by these authors is very interesting, but here are some limitations, in my opinion. The introduction is not very engaging, and the authors do not clearly present the limitations of the literature on the paper, nor do they address the gap that their work fills. There is no debate raised regarding the directions of the literature concerning new fields of research on the determinants.

If possible, please consider including a flow diagram for the sample screening process to facilitate better understanding of the sampling procedure. In terms of methodology, the authors do not explain the benefits of using a model and the gaps it addresses, as well as the potential robustness of the results compared to other studies.

Reviewer #2: Strengths:

The study employs an appropriate combination of cluster analysis and MANOVA to identify and validate student profiles based on motivation, procrastination, and adjustment.

The table clearly demonstrates meaningful differences between clusters, especially in terms of procrastination and intrinsic motivation.

The profiles generated align with GPA performance, making the results potentially useful for targeted educational interventions.

Suggestions for improvement:Please clarify how the number of clusters was determined (e.g., elbow method, silhouette analysis, or decision tree splits).

Reply: Thank you for pointing this out, see Appentix

It would strengthen the findings to elaborate on why extrinsic motivation shows no significant differences across profiles, and whether other subdimensions of motivation were considered.

Reviewer #3: The manuscript is generally well-executed, but I have the following suggestions:

Clarify Methodological Choices: It would be helpful to explain in more detail why the K-means and decision tree methods were selected over other potential techniques and the limitations associated with these methods. Contextualization of Findings:

Reply: Thank you for pointing this out, see Lines 219-232

Further exploration of how the results relate to the specific cultural and educational context in Greece would strengthen the manuscript, especially considering the unique influences of Greek educational and societal norms on student behavior.

Practical Applications: It would be beneficial to expand on how universities and educational programs can apply these findings in practical terms to support students better, focusing on interventions that target academic procrastination and motivation.

Consideration of Additional Variables: The study focuses on motivation, procrastination, and adjustment. Including additional variables such as socio-economic status or mental health might provide more comprehensive insights into student success.

Thank you so much for the suggestion, it would indeed be interesting to incorporate these variables in our future studies.

Reviewer #4: Higlights

This research is a great contribution to the field of educational psychology. Although the study focuses on variables that have been extensively studied such as academic adjustment, procrastination, and extrensic and intrinsic motivation, it provides a comprehensive discussion on the dynamic interplay of variables under study in Greeks’ context. Discussion of findings flows logically from the study results.

To enhance the paper for publication, the researchers may consider the following:

1. Provide additional studies in Greece on similar variables to illustrate the study’s context. Only one study in Greece focusing on the positive effect of academic adjustment on students’ academic success was included. It will enrich the study’s context if the researcher will provide additional studies along this topic.

Reply: Thank you for pointing this out, see Lines 46-56

2. Revise the introduction to improve its coherence and cohesiveness. The researcher may also include a brief background or context on the status of Greek college students’ academic adjustment, motivation and procrastination in general.

3. Include participants’ relevant demographics such as their year level in college and degree programs because these data can also provide insights to the study’s results.

Reply: Thank you for pointing this out, see Lines 142-152

4. Add the title, Results, after Statistical Analysis to provide demarcation between Methodology and Results Sections.

See line 238

5. Elaborate on specific interventions recommended in the conclusion section. Would it be school interventions only? Can policy makers be included too?

6. Ensure consistency in writing reference entries, capitalization of journal titles, use of period before the url. Observe hanging indentation.

7. The paper provides interesting, relevant, and insightful findings. To improve its clarity, consider improving awkward sentence construction and cohesion.

Reviewer #5: It is good that the manuscript recognizes the need for academia to remain diligent in seeking ways to continuously improve support of student outcomes.

However, in reading the manuscript I struggle to identify the meaningful contribution or expansion of knowledge. The constructs, academic motivation, procrastination, and adjustment have been extensively studied. Many of the findings are readily available and known

Perhaps a more beneficial study could assess the influence of online modalities or select AI policies in relation to academic motivation, procrastination, and adjustment.

---

## [Decision Letter · Decision Letter 1]

3 Aug 2025

Dear Dr. Kalamaras,

We look forward to receiving your revised manuscript.

Kind regards,

Assoc. Prof. Phakkharawat Sittiprapaporn, Ph.D.

Academic Editor

PLOS ONE

Journal Requirements:

Additional Editor Comments:

Reviewers' comments:

Reviewer's Responses to Questions

**Comments to the Author**

Reviewer #1: All comments have been addressed

Reviewer #2: All comments have been addressed

Reviewer #3: (No Response)

Reviewer #4: All comments have been addressed

2. Is the manuscript technically sound, and do the data support the conclusions?

Reviewer #1: Yes

Reviewer #2: Yes

Reviewer #3: Yes

Reviewer #4: Yes

3. Has the statistical analysis been performed appropriately and rigorously?

Reviewer #1: Yes

Reviewer #2: Yes

Reviewer #3: Yes

Reviewer #4: Yes

4. Have the authors made all data underlying the findings in their manuscript fully available?

Reviewer #1: Yes

Reviewer #2: Yes

Reviewer #3: Yes

Reviewer #4: Yes

5. Is the manuscript presented in an intelligible fashion and written in standard English?

Reviewer #1: Yes

Reviewer #2: Yes

Reviewer #3: Yes

Reviewer #4: Yes

Reviewer #1: (No Response)

Reviewer #2: Thank you for the opportunity to review the revised manuscript. The authors have carefully addressed the comments raised in the previous round, and the revisions have significantly improved the overall clarity and quality of the paper. The theoretical framework is now more coherent, the methodology is sound, and the discussion of the findings is insightful and well-supported by the data. I find the manuscript to be well-structured and of potential interest to the journal’s readership. I recommend that the paper be accepted for publication.

Reviewer #3: (No Response)

Reviewer #4: The study is a good addition to existing knowledge in the field of educational psychology that investigates the correlation among variables such as motivation, adjustment to college, and academic achievement.

The author was able to address the reviewers’ recommendations in the revised paper. The introduction incorporates studies in Greece on similar topic as well as research gaps. Ideas in the introduction flow naturally and cohesively. The methodology section is logically sound, clear, and replicable. The author also includes respondents’ relevant demographics and educational implications recommended by the reviewers.

However, I have three recommendations:

1. Include a graphic organizer (e.g. flow chart) for the study procedure from choosing the respondents to the statistical analysis to make it easy to grasp?

2. Provide a table for the respondents’ demographics to accompany the written description so that at glance, readers can make sense of it.

3. Add an introductory statement in the Results section before mentioning the findings of the study: As expected, procrastination assessment scale was negatively correlated to intrinsic motivation ...

Also, take note of the correct usage in the following sentences:

1. Freshmans make up 13.8% of the sample. (Correction: Freshmen)

2. The sample comprised of 36.2% male and 63.4% female students. (Correction: The sample comprised 36.2% male and ….)

**Do you want your identity to be public for this peer review?** For information about this choice, including consent withdrawal, please see our Privacy Policy

Reviewer #1: No

Reviewer #2: No

Reviewer #3: No

Reviewer #4: No

---

## [Author Response · Author response to Decision Letter 2]

4 Sep 2025

Comments to the Author

6. Review Comments to the Author

Reviewer #1: (No Response)

Reviewer #2: Thank you for the opportunity to review the revised manuscript. The authors have carefully addressed the comments raised in the previous round, and the revisions have significantly improved the overall clarity and quality of the paper. The theoretical framework is now more coherent, the methodology is sound, and the discussion of the findings is insightful and well-supported by the data. I find the manuscript to be well-structured and of potential interest to the journal’s readership. I recommend that the paper be accepted for publication.

Reviewer #3: (No Response)

Reviewer #4: The study is a good addition to existing knowledge in the field of educational psychology that investigates the correlation among variables such as motivation, adjustment to college, and academic achievement.

The author was able to address the reviewers’ recommendations in the revised paper. The introduction incorporates studies in Greece on similar topic as well as research gaps. Ideas in the introduction flow naturally and cohesively. The methodology section is logically sound, clear, and replicable. The author also includes respondents’ relevant demographics and educational implications recommended by the reviewers.

However, I have three recommendations:

1. Include a graphic organizer (e.g. flow chart) for the study procedure from choosing the respondents to the statistical analysis to ma ke it easy to grasp?

Reply: Thank you for pointing this out, see lines 181-183

2. Provide a table for the respondents’ demographics to accompany the written description so that at glance, readers can make sense of it.

Reply: Thank you for pointing this out, see lines 157-161

3. Add an introductory statement in the Results section before mentioning the findings of the study: As expected, procrastination assessment scale was negatively correlated to intrinsic motivation ...

Reply: Thank you for pointing this out, see lines 313-318

Also, take note of the correct usage in the following sentences:

1. Freshmans make up 13.8% of the sample. (Correction: Freshmen)

Reply: Thank you for pointing this out, see line 141

2. The sample comprised of 36.2% male and 63.4% female students. (Correction: The sample comprised 36.2% male and ….)

Reply: Thank you for pointing this out, see lines 138-139

Τhe remaining comments did not require any action from the authors

---

## [Decision Letter · Decision Letter 2]

18 Sep 2025

Dear Dr. Kalamaras,

Thank you for submitting your manuscript to PLOS ONE. After careful consideration, we feel that it has merit but does not fully meet PLOS ONE’s publication criteria as it currently stands. Therefore, we invite you to submit a revised version of the manuscript that addresses the points raised during the review process.

We look forward to receiving your revised manuscript.

Kind regards,

Assoc. Prof. Phakkharawat Sittiprapaporn, Ph.D.

Academic Editor

PLOS ONE

Journal Requirements:

Reviewer's Responses to Questions

**Comments to the Author**

Reviewer #4: All comments have been addressed

2. Is the manuscript technically sound, and do the data support the conclusions?

Reviewer #4: Yes

3. Has the statistical analysis been performed appropriately and rigorously?

Reviewer #4: Yes

4. Have the authors made all data underlying the findings in their manuscript fully available?

Reviewer #4: Yes

5. Is the manuscript presented in an intelligible fashion and written in standard English?

Reviewer #4: Yes

Reviewer #4: All comments have been addressed; however, there is one more that I missed out pointing. Apologies for this oversight. Kindly indicate in your Methodology your research design and purpose. Is it a correlational study? Is it a descriptive quantitative study? What is its aim?

**Do you want your identity to be public for this peer review?** For information about this choice, including consent withdrawal, please see our Privacy Policy

Reviewer #4: No

---

## [Author Response · Author response to Decision Letter 3]

25 Sep 2025

6. Review Comments to the Author

Reviewer #4: All comments have been addressed; however, there is one more that I missed out pointing. Apologies for this oversight. Kindly indicate in your Methodology your research design and purpose. Is it a correlational study? Is it a descriptive quantitative study? What is its aim?

Reply: Thank you for pointing this out, see lines 137-140

Τhe following comments of 4th reviewer did not require any action from the authors

---

## [Decision Letter · Decision Letter 3]

20 Oct 2025

Academic Motivation, Procrastination, and Adjustment: Exploring Their Impact on Student Profiles and Academic Performance

PONE-D-25-16926R3

Dear Dr. Kalamaras,

We’re pleased to inform you that your manuscript has been judged scientifically suitable for publication and will be formally accepted for publication once it meets all outstanding technical requirements.

Kind regards,

Assoc. Prof. Phakkharawat Sittiprapaporn, Ph.D.

Academic Editor

PLOS ONE

Additional Editor Comments (optional):

Reviewers' comments:

Reviewer's Responses to Questions

**Comments to the Author**

Reviewer #4: All comments have been addressed

2. Is the manuscript technically sound, and do the data support the conclusions?

Reviewer #4: Yes

3. Has the statistical analysis been performed appropriately and rigorously?

Reviewer #4: Yes

4. Have the authors made all data underlying the findings in their manuscript fully available?

Reviewer #4: Yes

5. Is the manuscript presented in an intelligible fashion and written in standard English?

Reviewer #4: Yes

Reviewer #4: In line 112, concept is mispelled. Kindly correct it.

in line 246, is it Result or Results? Please check because what I know is the section title is Results.

**Do you want your identity to be public for this peer review?** For information about this choice, including consent withdrawal, please see our Privacy Policy

Reviewer #4: No

---

## [Editor Report · Acceptance letter]

PONE-D-25-16926R3

PLOS ONE

Dear Dr. Kalamaras,

I'm pleased to inform you that your manuscript has been deemed suitable for publication in PLOS ONE. Congratulations! Your manuscript is now being handed over to our production team.

Kind regards,

on behalf of

Assoc. Prof. Dr. Phakkharawat Sittiprapaporn

Academic Editor

PLOS ONE